# Effects of Wet Fermented Soybean Dregs on Physical and Mechanical Properties of Pellets of Corn Stover

**DOI:** 10.3390/ani12192632

**Published:** 2022-09-30

**Authors:** Tianyou Chen, Wenyu Zhang, Yuxin Liu, Yuqiu Song, Liyan Wu, Cuihong Liu, Tieliang Wang

**Affiliations:** 1College of Engineering, Shenyang Agricultural University, Shenyang 110866, China; 2College of Water Conservancy, Shenyang Agricultural University, Shenyang 110866, China

**Keywords:** fermented soybean dregs, corn stover, pelleting, physical and mechanical properties

## Abstract

**Simple Summary:**

Corn stover pellets are of poor quality (regarding physical and mechanical properties), and wet soybean dregs are viscous, which may improve the pelleting process. Furthermore, the soybean dregs have high nutritional value, which may increase the quality of feed. The relaxed density, dimensional stability coefficient and hardness of feed pellets were measured and analyzed in this study to clarify the influence mechanism of wet-fermented soybean dregs on the pelleting of corn stover. The results showed that fermented soybean dregs had a significant effect on corn stover pelleting. The comprehensive analysis showed that the addition of 5–10% fermented soybean dregs to corn stover improved the relaxed density, dimensional stability coefficient, and hardness of feed pellets by 10.76–23.51%, 7.32–15.74%, and 33.39–454.47%, respectively. The results could provide guidance on the research and development of high-quality forming technology and instruments using the mixture of corn stover and fermented soybean dregs and offer a new route for the utilization of high-moisture materials.

**Abstract:**

Pelleting experiments were carried out in this study to clarify the influence mechanism of wet-fermented soybean dregs on the forming of corn stover. The effects of addition of water or wet fermented soybean dregs on the forming quality of corn stover were comparatively studied under different corn stover particle sizes and compression displacement. The fermented soybean dregs significantly affected the relaxed density, dimensional stability coefficient, and hardness of feed pellets. The relaxed density, dimensional stability coefficient, and hardness of feed pellets increased first and then decreased with the increase of fermented soybean dregs. The forming quality of corn stover added with fermented soybean dregs was higher than that of corn stover added with the same amount of water. The mechanism allowed soybean dregs to strengthen the bonding between corn stover particles and thus improved the quality of feed pellets. A certain amount of water was favorable for corn stover pelleting, but excessive water may decrease the quality of pellets. The comprehensive analysis showed that the addition of 5–10% fermented soybean dregs to the corn stover improved the relaxed density, dimensional stability coefficient, and hardness of feed pellets by 10.76–23.51%, 7.32–15.74%, and 33.39–454.47%, respectively.

## 1. Introduction

Crop straws are renewable resources, with an annual yield over 2 billion tons in the world and 0.9 billion tons in China [1]. Corn stover comprises about one-third of all crop straws [2]. Corn stover is rich in crude fibers and proteins, which can be utilized efficiently in ruminant production [3]. There are different patterns for the use of corn stover as livestock feed, including fermenting, pressing block, pelleting, and puffing [4]. The pellets can increase the feed intake, digestibility, and daily gain of animals, and can be easily transported and stored, effectively solving problems such as the seasonal and annual imbalance of feed supply and demand, high transportation and storage cost, and long feeding time and high labor intensity [5,6]. The physical and mechanical properties are the evaluation indexes of the quality of forming materials [7]. However, the molding quality of corn stover is poor and the utilization rate is still low [8]. Therefore, seeking effective additives is necessary to improve corn stover forming quality.

The effects of various additives on the forming quality of straws have been studied. Additives include corn starch, beet, cassava starch, calcium carbonate, apple pomace, vegetable pomace, and potato residues. Corn starch or beet can improve the durability of alfalfa, wheat, and rape straw pellets [7]. When 4% cassava starch or calcium carbonate was added to wheat straw, the density of straw pellets was the highest [9]. Apple and vegetable pomace can significantly improve the hardness and density of barley straw pellets [10]. The addition of potato residues improved the straw forming and decreased the density relaxation ratio of pellets [11]. During material forming, moisture content is a key influence factor on straw forming [12,13]. Reportedly, the forming performance index of corn stover of 5–8 mm was optimal when the moisture content was 8–12% [14]. In the water content range of 8–24% for corn stover powder under mesh of 16–100, the relaxation density of the formed pellets decreased with an increase in water content [15]. The optimal forming moisture content was 13.63% for chopped corn stover [16]. The forming experiments of unbundled corn stover showed that the relaxed density of straw briquette increased first and then dropped with the increase in moisture content, and the optimal moisture content was 15% [17]. These findings indicate that the optimal moisture content is favorable for pelleting and differs under different characteristics of corn stover.

Soybean dregs, a byproduct from tofu, soybean milk, and sufu processing, are viscous with a high moisture content (>75%) and may improve the pelleting process and quality. Furthermore, Soybean dregs are of high nutritional value, owing to the abundant dietary fibers, proteins, lipids, vitamins and minerals [18,19] Therefore, the addition of soybean dregs provide the possibility for high quality molding and the utilization of corn stover. However, soybean dregs are prone to rotting and deterioration and cannot be stored in the long term. They contain anti-nutrient factors, including trypsin inhibitor, phytic acid, and tannin. Among these, the trypsin inhibitor factor can hinder the digestion and absorption of legume protein in the animal body, causing diarrhea and affecting the growth of animals [20]. Fermentation can effectively reduce the anti-nutritive factors in soybean dregs [21]. Total mixed fermentation with soybean dregs may help improve growth performance in growing Hanwoo heifers [22]. When fermented soybean dregs were used to replace soybean meal in the diet, 5% fermented soybean dregs significantly improved the feed–weight rate and weight gain benefit of beef cattle [23]. Reportedly, okara is an effective source of proteins for lactating ewes and their twin lambs [24]. Zang et al. [25] reported that okara meal can completely replace soybean meal without negatively affecting production and nutrient digestibility in early- to mid- lactation Jersey cows. Rahman et al. [26] reported that the goats fed with supplemental soybean waste had a lower total dry matter intake, feed conversion ratio, and feed cost per kilogram of body weight gain than those fed with commercial pellets. In sum, fermented soybean dregs can be easily digested and absorbed by animals and are a substitute of soybean meal and other protein feeds. However, inappropriate storage will induce mildewing [27,28], and only 10% of soybean dregs can be processed and used, which is a severe waste of resources [29,30].

Therefore, the effects of the addition of wet-fermented soybean dregs on the forming quality of corn stover were comparatively investigated in this study to clarify the mechanism of fermented soybean dregs on corn stover pelleting. 

## 2. Materials and Methods

### 2.1. Materials

Fresh soybean dregs were bought from a tofu processing factory of Shenyang Agricultural University and added along with effective microorganism yeast starters (Wangnongbao Biotechnology Co., Ltd., Zhengzhou, China). After 10 d of fermentation, the wet base (w.b.) moisture content of soybean dregs was 80%. The corn stover (variety: Suyu 28) was chopped with a hammer mill, and then screened using 2 mm and 4 mm sieves to obtain two kinds of materials of particle size [31]. The moisture content of corn stover was 8% (w.b.). Table 1 showed the nutrient components of corn stover and fermented soybean dregs.

### 2.2. Methods

#### 2.2.1. Materials Treatments and Pelleting

The wet fermented soybean dregs are made up of dry matter and water. Investigations were carried out to compare the effects of the addition of fermented soybean dregs and water on corn stover pelleting and to explore the influence mechanism of fermented soybean dregs on the molding under different parameters. The corn stover samples of 4 g (8%, w.b.) at size of 2 mm and 4 mm were weighed with electronic scales (BS200S, Sartorius, Gottingen, Germany) and sealed in bags. The fermented soybean dregs of 0.2, 0.4, 0.6, 0.8, or 1.0 g were weighed and evenly added to the corn stover samples of 2 and 4 mm separately, and then blended and placed in sealed bags to obtain the mixed feed samples (the added levels of fermented soybean dregs were 5%, 10%, 15%, 20%, and 25%, respectively). The levels of soybean dregs tested were based on previous reports, in which the tested inclusion level of soybean residues to ruminant diets was from 5% to 24% without negative effects on DM intake or productivity [32,33,34,35]. Thus, it can be expected that the additional mass of soybean dregs tested would not significantly affect the needs of animals fed with the soybean dregs–corn stover pellets under the better pellet forming conditions. The moisture content of the comparison samples was adjusted according to that of samples added with fermented soybean dregs (0.15, 0.31, 0.45, 0.60, and 0.74 g, respectively). The samples were sealed and stored at room temperature for 48 h to allow the water to evenly distribute.

The feed forming testing system consisted of a forming mould (internal diameter of 20 mm, height of 103.5 mm), pressure head, blocking plates, a WDW-200 microcomputer controlled electronic universal testing machine (Jinan Shijin Group Co., Ltd., Jinan, China), and a base. The compression displacement of the pelleting of corn stover and mixed feed was set at 90 mm and 92 mm, respectively, according to the preliminary experiments. In the feed forming process, the materials were added to the mould, and then the universal testing machine was started and operated at the rate of 100 mm/min. When the set compression displacement was reached, the machine was held for 30 s [36]. Then the pressure head was quickly removed manually, and the formed pellets were pushed out at the rate of 180 mm/min. Finally, the indenter returned to the initial position.

#### 2.2.2. Indices and Detection Methods

(1)Relaxed Density

Since corn stover is viscoelastic, the compressed pellets during storage will slowly expand in size until stabilization, where the density is called the relaxed density. The dimensional change and relaxation density of pellets directly affect the transportation cost and feed storage, which are important indices for pelleting. Preliminary experiments showed that the feed pellet size stabilized after 7 d of storage. For this reason, the feed pellet sizes were measured after 7 d of storage. Then relaxed density (RDS, kg/m^3^) was calculated as follows [37]: (1)$$RDS=\frac{4m\times 10^{9}}{\pi \cdot D^{2}\cdot H}$$
where *m* is the mass of pellets, kg, and *D* and *H* are the diameter and height of pellets after storage of 7 d, respectively, mm. 

(2)Dimensional Stability Coefficient

The pellet diameter and height were measured and volume was calculated after feed forming and after storage of 7 d. The dimensional stability coefficient of feed pellets (*DS*, %) was computed as follows [38]:(2)$$DS=(1-\frac{V_{t}-V_{0}}{V_{0}})\times 100\%$$
where *V_t_* and *V*_0_ are the pellet volumes after storage of 7 d and immediately after forming, respectively, m^3^. 

(3)Hardness

Hardness is an important index of the appearance quality of pelleted feed and has a certain effect on production performance of animals [39]. Therefore, the feed pellets after storage of 7 d were tested under compression on a TMS-Pro texture analyzer (Beijing Ensoul Technology Ltd., Beijing, China) at the compression speed of 10 mm/min. The forces during the destruction were recorded, and the largest force was exactly the hardness of feed pellets [40].

#### 2.2.3. Data Analysis

SPSS 26.0 (SPSS Inc., Chicago, IL, USA) was used to analyze one-way ANOVA and determine the significance of differences between mean values by Tukey HSD (*p* < 0.05). The results were presented as mean ± SD. Origin 8.0 (Origin Lab, Northampton, MA, USA) was employed to plot diagrams. 

## 3. Results and Discussion

### 3.1. Relaxed Density of Feed Pellets

#### 3.1.1. Effects of Moisture Content on Relaxed Density of Corn Stover Pellets (CSP)

The effects of moisture content on the relaxed density of CSP are illustrated in Figure 1. For the pellets from corn stover of 2 mm, the relaxed density of the CSP increased first and then declined with the increase of water addition. When the water addition was 0.15 g (at moisture content of 11.33%), the relaxed densities of CSP were the greatest at compression displacements of 90 mm and 92 mm separately, which increased by 10.37% and 7.62%, respectively, relative to that of corn stover without water addition. For the pellets from corn stover of 4 mm, the relaxed density of CSP decreased with the increase of water addition. The reasons for the above phenomena were that the elasticity of molding materials increased, and the bonding force weakened when the moisture content exceeded the optimal level, so the relaxed density of pellets decreased. These conclusions are consistent with previous studies [41,42].

For the CSP from the same particle size of corn stover, the relaxed density of the CSP increased with the rise of compression displacement. Reportedly, the pressure was significant on density, and the density of formed pellets increased with the increase in pressure and was relatively stable after a certain pressure [43,44]. Compared with the feed pellets from corn stover of 4 mm, the relaxed density of CSP from corn stover of 2 mm was larger at a low moisture content but was smaller at a high moisture content. The reasons for the above phenomena might be that small-size biomass particles easily fill in the gaps between them, leading to a smaller pellet volume [45]. When the moisture content reaches a certain level, the elasticity of materials increased, and the bonding force is weakened. While the bonding area between large sized straws became larger, and the resilience was smaller. 

#### 3.1.2. Effects of Wet Fermented Soybean Dregs on Relaxed Density of Corn Stover Pellet with Wet Soybean Dregs Addition (CSPSD)

The relaxed density of CSPSD with different masses of fermented soybean dregs and of CSP without water addition were illustrated in Figure 2. The relaxed density of CSPSD increased first and then declined with the increase of fermented soybean dregs, indicating the optimal addition of soybean dregs. For the pellets from corn stover of 2 mm, the relaxed densities of CSPSD were the greatest at compression displacements of 90 mm and 92 mm separately when the addition–mass ratio of fermented soybean dregs was 5%, and they increased by 23.51% and 18.23%, respectively, relative to that of corn stover without the addition of fermented soybean dregs. For the pellets from corn stover of 4 mm, the relaxed density of CSPSD was the greatest at compression displacements of 90 mm and 92 mm separately when the addition–mass ratio of fermented soybean dregs was 10% and they increased by 18.54% and 14.93%, respectively, relative to that of corn stover without the addition of fermented soybean dregs. 

At the same moisture content, the relaxed densities of CSPSD were larger than CSP (Figure 1 and Figure 2). In conclusion, the soybean dregs can enhance the bonding between corn stover particles and improve the relaxed density of the formed pellets. However, when the moisture content in materials exceeded a certain level, the relaxed density of the pellets decreases [16], which also explains why the relaxed density declines after the addition of fermented soybean dregs was above a certain level. Compared with the feed pellets from the corn stover of 4 mm, the relaxed density of formed pellets from corn stover of 2 mm was larger, and the difference between the two became smaller when the addition–mass ratio of fermented soybean dregs increased (>5%). The effect of soybean dregs was more significant on the bonding force between small-size corn stover particles. When the compression displacement increased, the relaxed density of feed pellets increased, which was similar to the effect on corn stover forming. The addition of fermented soybean dregs at 5–10% can increase the relaxed density by 10.76–23.51% compared with that of pellets from only corn stover. 

### 3.2. Dimensional Stability Coefficient of Feed Pellets

#### 3.2.1. Effects of Moisture Content on Dimensional Stability Coefficient of CSP

The effects of moisture content on the dimensional stability coefficient of CSP were illustrated in Figure 3. The dimensional stability coefficient of CSP increased first and then decreased with the increase of moisture content. For the pellets from corn stover of 2 mm, when the compression displacement was 90 mm and the water addition was 0.15 g (at moisture content of 11.33%), the dimensional stability coefficient of feed pellets was the greatest and it was 94.37%. When the compression displacement was 92 mm and the water addition was 0.31 g (at moisture content of 14.62%), the dimensional stability coefficient of CSP was the greatest at 96.81%, and it increased by 11.88% relative to that of corn stover without water addition. For the 4 mm corn stover, when the compression displacement was 90 mm and the water addition was 0.31 g (at moisture content of 14.62%), the dimensional stability coefficient of CSP was the greatest and it was 87.9%, and it increased by 6.17% relative to that of corn stover without water addition. When the compression displacement was 92 mm and the water addition was 0.6 g (at moisture content of 20%), the dimensional stability coefficient of CSP was the greatest and it was 90.29%. The reason for the above phenomena was that the appropriate moisture content was conducive to corn stover forming and increased forming density, but when the moisture content exceeded a certain level, the corn stover elasticity increased and the density of the formed pellet decreased. Moreover, the water in feed pellets at high level evaporated during storage, leading to a weaker bonding force and larger pellet size. 

Compared with the pellets from corn stover of 4 mm in particle size, the dimensional stability coefficient of the pellets from corn stover of 2 mm was always larger (except at the water addition of 0.6 g and compression displacement of 92 mm). This was because the bonding between smaller-size straw particles was tighter, and the volume of formed pellets was more stable. The compression displacement did not significantly affect the dimensional stability coefficient of CSP. This was because the dimensional stability coefficient was determined jointly by the forming density and relaxed density, but when the compression displacement increased, both the forming density and relaxed density increased. 

#### 3.2.2. Effects of Wet Fermented Soybean Dregs on Dimensional Stability Coefficient of CSPSD

The dimensional stability coefficient of CSPSD with different masses of fermented soybean dregs and CSP without water addition were illustrated in Figure 4. The dimensional stability coefficient of CSP increased first and then decreased with the increase of fermented soybean dregs. The dimensional stability coefficient of CSPSD was higher than that of corn stover without addition. For the feed pellets from corn stover of 2 mm, at the compression displacement of 90 mm, the dimensional stability coefficient of CSPSD was the greatest when the addition–mass ratio was 5%, and it increased by 9.5% relative to that of corn stover without the addition of fermented soybean dregs. At the compression displacement of 92 mm, the dimensional stability coefficient of CSPSD was the greatest when the addition–mass ratio was 10%, it increased by 12.61% relative to that of corn stover without the addition of fermented soybean dregs. For the feed pellets from corn stover of 4 mm, the dimensional stability coefficients of CSPSD were the greatest at compression displacements of 90 mm and 92 mm separately when the addition–mass ratio was 10%, and they increased by 15.74% and 14.65%, respectively, relative to that of corn stover without the addition of fermented soybean dregs.

At the same moisture content, the dimensional stability coefficients of CSPSD were larger than those of CSP (Figure 3 and Figure 4), indicating the soybean dregs can stabilize the sizes of feed pellets. The effects of corn stover particles and compression displacement on the dimensional stability coefficient of CSPSD were the same as CSP, which was because the mixed feeds were mainly composed of corn stover. The addition of fermented soybean dregs at 5–10% can increase the dimensional stability coefficient by 7.32–15.74% compared with that of pellets from only corn stover. 

### 3.3. Hardness of Feed Pellets

#### 3.3.1. Effects of Moisture Content on Hardness of CSP

The effects of moisture content on the hardness of CSP were illustrated in Figure 5. The hardness of CSP increased first and then declined with the increase of water addition. For the pellets from corn stover of 2 mm, when the water addition was 0.45 g (at moisture content of 17.3%), the hardness of CSP was the greatest at compression displacements of 90 mm and 92 mm separately, and they increased by 468.02% and 86.97%, respectively, relative to that of corn stover without the addition of water. For the pellets from the corn stover of 4 mm, when the compression displacement was 90 mm and the water addition was 0.45 g (at moisture content of 17.3%), the hardness of CSP was the greatest, and it increased by 90.62% relative to that of the corn stover without water addition. When the compression displacement was 92 mm and the water addition mass was 0.6 g (at moisture content of 20%), the hardness of CSP was the greatest, and it increased by 65.71% relative to that of corn stover without the water addition. The reason for the above phenomena was that moisture can improve the bonding strength between the corn stover particles, but when the moisture content was too high, the relaxed density of pellets decreased and the strength was weakened accordingly. Huang et al. [46] found that the hardness of pellets increased with the increase of moisture content and decreased after reaching the optimal moisture content. 

Compared with the pellets of corn stover of 2 mm, the hardness of the pellets from corn stover of 4 mm was larger. These reason was that the Van der Waals force between pellets enhanced with the increase of material particles size, so the mechanical strength increased [47]. Dai et al. [48] found that both durability and shatter resistance rose first and then declined, with increase of the material particles, and the differences may be attributed to the different ranges of straw particles size. At the same particle size of corn stover, when the compression displacement rose, the hardness of feed pellets increased. The reasons were that when the compression force increased, the mechanical bonding between corn stover particles was tighter, and the mechanical strength was improved. Jiang et al. [49] found that the hardness of biomass pellets increased with the increment of pressure.

#### 3.3.2. Effects of Wet Fermented Soybean Dregs on Hardness of CSPSD

The hardness of CSPSD with different masses of fermented soybean dregs and CSP without water addition were illustrated in Figure 6. The hardness of CSPSD rose first and then decreased with the increase of fermented soybean dregs, and it was higher in comparison with the corn stover without addition of soybean dregs. At the compression displacement of 90 mm, the hardness of feed pellets from corn stover of 2 mm and 4 mm was the greatest when the addition–mass ratio of fermented soybean dregs was 15%, and increased by 509.44% and 128.03%, respectively, compared with the pellets from corn stover without addition. At the compression displacement of 92 mm, the hardness of feed pellets was the greatest when the addition–mass ratio of fermented soybean dregs was 10% and increased by 115.45% and 89.08%, respectively, compared with the pellets from corn stover without addition. The reason for the above phenomena was that the fermented soybean dregs can enhance the bonding between corn stover particles, but when the addition was over a certain level, the hardness of CSPSD decreased, since the increase in moisture content and the decrease in forming density and the inter-pellet cohesive force. Celma et al. [50] reported that the hardness of pellets from industrial tomato residues was up to 88 N. Zhang et al. [51] obtained that the hardness of feed pellets was 450 N through optimization method.

At the same moisture content, the hardness of CSPSD was larger than that of CSP (Figure 5 and Figure 6). This was because the soybean dregs can increase inter-pellet bonding, the relaxed density, and dimensional stability coefficient of CSPSD, so the hardness increased accordingly. In the mixed feeds, the effects of corn stover particles size and compression displacement on the hardness of feed pellets were the same as those in the corn stover, because the mixed feeds were mainly composed of corn stover. The addition of fermented soybean dregs at 5–15% can increase the hardness by 33.39–509.44% compared with that of pellets from only corn stover. Comprehensive analysis showed that the addition of fermented soybean dregs by 5–10% to the corn stover can improve the hardness of feed pellets by 33.39–454.47%.

## 4. Conclusions

The fermented soybean dregs and moisture content affected the relaxed density, dimensional stability coefficient and hardness of feed pellets, which all increased first and then decreased with the increase of fermented soybean dregs. The soybean dregs can enhance the bond between corn stover particles and improve the quality of feed pellets. A suitable amount of water was conducive to the molding of corn stover, and the molding quality declined at a moisture content of over a certain level. When the moisture content of the materials was low (<17.3%), the relaxed density and dimensional stability coefficient of pellets from the corn stover of smaller particle size was larger, while the hardness of pellets was lower.

## Figures and Tables

**Figure 1 animals-12-02632-f001:**
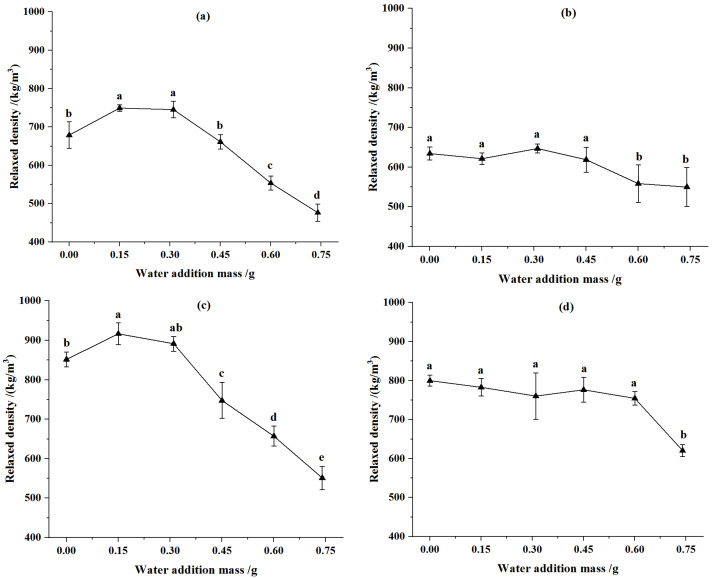
Effect of water addition on relaxed density of pellets. (**a**) Particle size was 2 mm and compression displacement was 90 mm. (**b**) Particle size was 4 mm and compression displacement was 90 mm. (**c**) Particle size was 2 mm and compression displacement was 92 mm. (**d**) Particle size was 4 mm and compression displacement was 92 mm. Note: Different letters in the same line chart indicate a significant difference between the data (*p* < 0.05).

**Figure 2 animals-12-02632-f002:**
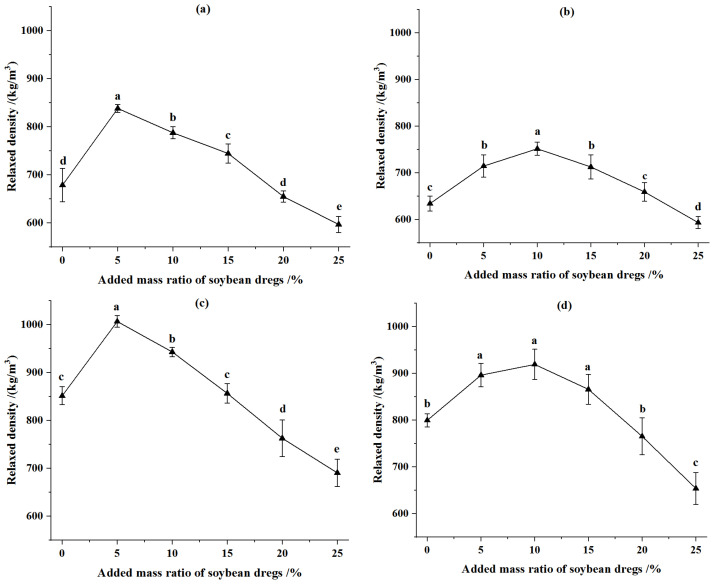
Effect of added mass ratio of fermented soybean dregs on relaxed density of pellets. (**a**) Particle size was 2 mm and compression displacement was 90 mm. (**b**) Particle size was 4 mm and compression displacement was 90 mm. (**c**) Particle size was 2 mm and compression displacement was 92 mm. (**d**) Particle size was 4 mm and compression displacement was 92 mm. Note: Different letters in the same line chart indicate a significant difference between the data (*p* < 0.05).

**Figure 3 animals-12-02632-f003:**
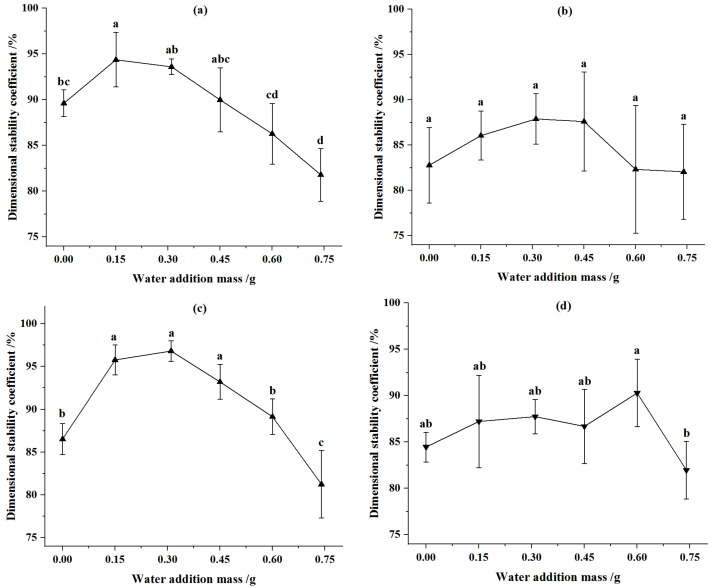
Effect of water addition on dimensional stability coefficient of pellets. (**a**) Particle size was 2 mm and compression displacement was 90 mm. (**b**) Particle size was 4 mm and compression displacement was 90 mm. (**c**) Particle size was 2 mm and compression displacement was 92 mm. (**d**) Particle size was 4 mm and compression displacement was 92 mm. Note: Different letters in the same line chart indicate a significant difference between the data (*p* < 0.05).

**Figure 4 animals-12-02632-f004:**
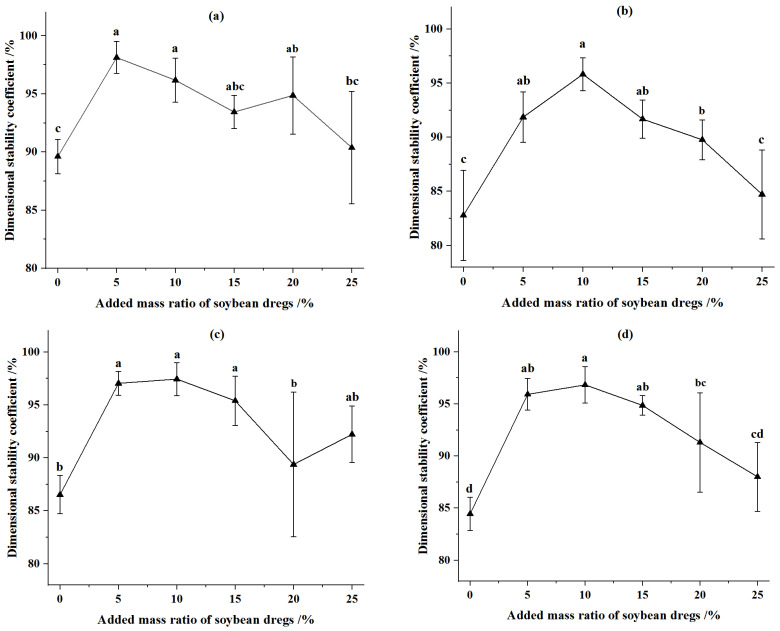
Effect of added mass ratio of fermented soybean dregs on dimensional stability coefficient of pellets. (**a**) Particle size was 2 mm and compression displacement was 90 mm. (**b**) Particle size was 4 mm and compression displacement was 90 mm. (**c**) Particle size was 2 mm and compression displacement was 92 mm. (**d**) Particle size was 4 mm and compression displacement was 92 mm. Note: Different letters in the same line chart indicate a significant difference between the data (*p* < 0.05).

**Figure 5 animals-12-02632-f005:**
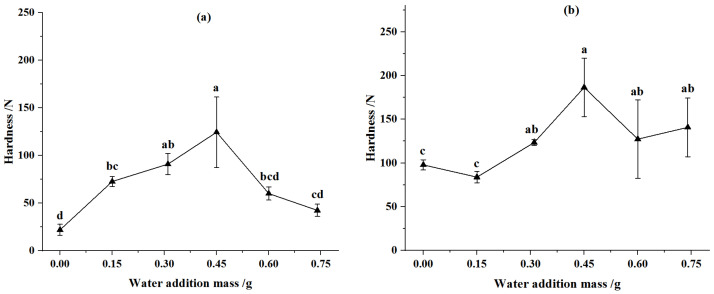
Effect of water addition on hardness of pellets. (**a**) Particle size was 2 mm and compression displacement was 90 mm. (**b**) Particle size was 4 mm and compression displacement was 90 mm. (**c**) Particle size was 2 mm and compression displacement was 92 mm. (**d**) Particle size was 4 mm and compression displacement was 92 mm. Note: Different letters in the same line chart indicate a significant difference between the data (*p* < 0.05).

**Figure 6 animals-12-02632-f006:**
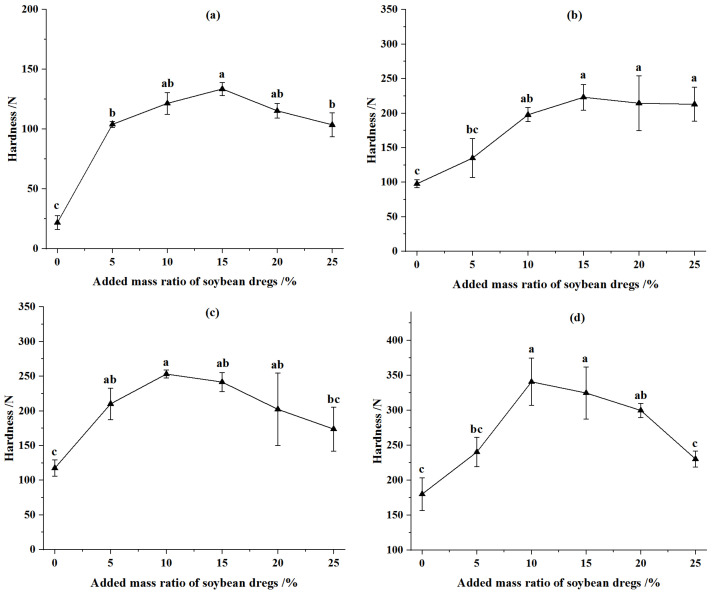
Effect of added mass ratio of fermented soybean dregs on hardness of pellets. (**a**) Particle size was 2 mm and compression displacement was 90 mm. (**b**) Particle size was 4 mm and compression displacement was 90 mm. (**c**) Particle size was 2 mm and compression displacement was 92 mm. (**d**) Particle size was 4 mm and compression displacement was 92 mm. Note: Different letters in the same line chart indicate a significant difference between the data (*p* < 0.05).

**Table 1 animals-12-02632-t001:** Nutrient components of corn stover and fermented soybean dregs (air dried) ^a^.

Materials	Dry Matter	Organic Matter	Crude Protein	Crude Fat	Neutral Detergent Fiber	Acid Detergent Fiber
Corn stover	92.02	91.12	7.02	1.25	68.24	39.51
Fermented soybean dregs	91.86	88.22	17.23	4.85	36.15	17.78

Note: ^a^ indicated that the unit of nutriment components was %.

## Data Availability

The datasets analyzed in the current study are available from the lead author on reasonable request.

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
