# Peer review of "Effects of Wet Fermented Soybean Dregs on Physical and Mechanical Properties of Pellets of Corn Stover"

_animals, 2022, doi:10.3390/ani12192632_

Round 1
Reviewer 1 Report
The topic, the use of by-products (wet fermented soybean dregs) to improve feed quality (in this case corn stover pellets) is an important topic noadays. However the current paper has important deficiencies that difficults it's reading.
The main concerns are:
Title: should be rewritten, indicating either the aim of the study or a result..
Introduction: which is really the aim of the study? which is the hypothesis? adding wet soybean dregs because of its water content may help to obtain a better corn stover pellet quality?
Throughout the paper: forming quality ...what does it really mean? the quality after the pellet production? at the day of manufacture...say it simple, say the quality of the pellet at the day of manufacture...
Which are the treatments really? feed pellets that usually are only contain corn stover? I am confused throughout the paper, which is the difference among straw pellets (contain only corn stover) and feed pellets? are corn stover pellets not used to feed the animals? Different particle sizes and compression displacement were evaluated...so was it a factorial design of the treatments? where are the interactions in the statistical model...
Line 109 to 122 I tried to understand the treatments...but I still do not..line 109 only 4 g of corn stover are used?
Why is relaxed density for animal nutrient use? in the materials and methods section I miss an explanation that describes or indicates the importance of the data recorded in relation to eating behaviour, nutrient availability...
Is there no nutritional analyses? DM, OM, CP...?
There are other important issues to be corrected like...what does the figure 2 inform us? I cannot differentiate the treatments...
I stopped here- I need a focused introduction, that indicates me the importance of all parameters tested, with an hypothesis...I need a proper treatment description, if it's a factorial design of treatments, I need to see the interactions...I need to understand why the measurements are important in relation to animal nutrition ( it may only be related to feed storage, which is also important), to be easy to read- it should be simple (either straw pellet or corn stover pellet and corn stover pellet with wet soybean dregs addition)...
I think the paper need a deep review before we can continue reviewing it. I encourage the authors to think about it, there is a lot of work behind this paper, however the current submission does not allow to follow it properly.
Author Response
Dear Reviewer: Thank you for your comments concerning our manuscript entitled “Wet Fermented Soybean Dregs and Corn Stover Pelleting for Ruminants Feeds” (ID: animals-1788697). We quite appreciate your favorite consideration and the your insightful comments. We did a major revision to the manuscript according to your helpful comments. We hope this revision can make our manuscript more acceptable. Revised portion are marked in blue in the manuscript. The revisions were addressed point by point below. Please feel free to contact us if there are any questions. Sincerely yours, Response to the reviewer’s comments: The topic, the use of by-products (wet fermented soybean dregs) to improve feed quality (in this case corn stover pellets) is an important topic nowadays. However the current paper has important deficiencies that difficults it's reading. The main concerns are: 1. Title: should be rewritten, indicating either the aim of the study or a result. Response: Thank you for your comments. We revised the title of the manuscript into “Effects of Wet Fermented Soybean Dregs on Physical and Mechanical Properties of Pellets of Corn Stover” according to your and other reviewer’s valuable suggestion. 2. Introduction: which is really the aim of the study? which is the hypothesis? adding wet soybean dregs because of its water content may help to obtain a better corn stover pellet quality? Response: Thank you for your comments. The soybean dregs have high nutritional values owing to the abundant dietary fibers, proteins, lipids, vita-mins and minerals [1-2]. The pellet feed can increase feed intake, digestibility and daily gain of animals, and it can be easily transported and stored. It can effectively solve the problems such as seasonal and annual supply and demand imbalance of feed, the high transportation and storage cost of corn stover [3-4]. Wet fermented soybean dregs contain dry matter and water, and the pelleting characteristics of mixture of wet fermented bean dregs and corn stover is not clear. Therefore, the purpose of this paper is to explore the effect of wet fermented soybean dregs (soybean dregs and water) on the relaxation density, dimensional stability and hardness of corn stover pellets. Ref: [1] Zhao, C.; Ma, G.; Lv, J.; Jiang, X.; Zhang G. Effects of Adding Lactic Acid Bacteria and Cellulase on Quality of Mixed Silage of Soybean Residue and Mulberry Leaves and Rumen Fermentation Characteristics in Vitro. Chinese Journal of Animal Nutrition 2021, 33, 2168-2177. doi:10.3969/j.issn.1006-267x.2021.04.036. (In Chinese with English Abstract). [2] Guo, J.; Dou, K.; Wang, F.; Wang, D. Application progress of soybean dregs in food. Cereals & Oils 2022, 35, 15-17. (In Chinese with English Abstract). [3] Hong, J.; Wang, Y.; Zhao, F.; Liu, X.; Quan, J.; Niu, X.; Han, X. Effects of straw pellets on rumen function and live weight gain of beef cattle. Pratacultural Science 2016, 25, 163-170. doi:10.11686/cyxb2016129. (In Chinese with English Abstract). [4] Zhao, F.; Zhang, H.; Hong, J.; Quang, J.; Li S.; Zhao Z.; Zuo Z. Effects of replacement of whole corn silage with straw pellets on finishing steers. Pratacultural Science 2020, 37, 2089-2096. doi:10.11829/j.issn.1001-0629.2019-0612. (In Chinese with English Abstract). 3. Throughout the paper: forming quality ...what does it really mean? the quality after the pellet production? at the day of manufacture...say it simple, say the quality of the pellet at the day of manufacture... Response: Thank you for your comments. Actually, the “forming quality” should be the physical and mechanical properties of pellets immediately after production and after stored for 7 days. The explanation was added in the manuscript. 4. Which are the treatments really? feed pellets that usually are only contain corn stover? I am confused throughout the paper, which is the difference among straw pellets (contain only corn stover) and feed pellets? are corn stover pellets not used to feed the animals? Different particle sizes and compression displacement were evaluated...so was it a factorial design of the treatments? where are the interactions in the statistical model... Response: Thank you very much for your valuable comments. We are very sorry for that the description of the experimental treatments was not clear. The treatments should be the addition of soybean dregs or water to the chopped corn stover of 2 mm and 4 mm, and pelleting at compression displacement of 90 mm and 92 mm, and pure corn stover(without addition of the dregs or water) was taken as the control. This study was to explore the effect mechanism of fermented soybean dregs on corn stover pelleting. The principal factors are the addition amount or ratio of fermented soybean dregs or water. The particle size and compression displacement are conditions for the pelleting, the results under these conditions are compared, but the experimental plan was not for the study of these conditions and their interactions. The corn stover pellets was called CSP for short, and corn stover pellets with wet soybean dregs addition was called CSPSD for short according to your suggestion. The related terms were revised in the manuscript. 5. Line 109 to 122 I tried to understand the treatments...but I still do not. line 109 only 4 g of corn stover are used? Response: Thank you for your comments. We are very sorry that our description of the trial plan is not clear, resulting in poor readability. According to your suggestion, we revised the description in the manuscript. The weight of corn stover for each pellet was 4 g in this study. 6. Why is relaxed density for animal nutrient use? in the materials and methods section I miss an explanation that describes or indicates the importance of the data recorded in relation to eating behaviour, nutrient availability... Response: Thank you for your valuable comments. The relaxed density is closely related to the hardness which is important for feed of animals, and the dimensional change and relaxation density of pellets directly affect the transportation and storage of feed. The description of the importance of the indices were added in the manuscript. 7. Is there no nutritional analyses? DM, OM, CP...? Response: Thank you for your valuable comments. The nutrient composition of corn stover and fermented soybean dregs were added in the manuscript according to your comments. 8. There are other important issues to be corrected like...what does the figure 2 inform us? I cannot differentiate the treatments... Response: Thank you for your comments. We checked carefully the manuscript and made major revisions. In fact, Figure 2 showed only part of the feed pellets and was meaningless. Therefore, figure 2 has been removed according to your and another reviewer’s comments. 9. I stopped here- I need a focused introduction, that indicates me the importance of all parameters tested, with an hypothesis...I need a proper treatment description, if it's a factorial design of treatments, I need to see the interactions...I need to understand why the measurements are important in relation to animal nutrition ( it may only be related to feed storage, which is also important), to be easy to read- it should be simple (either straw pellet or corn stover pellet and corn stover pellet with wet soybean dregs addition)... I think the paper need a deep review before we can continue reviewing it. I encourage the authors to think about it, there is a lot of work behind this paper, however the current submission does not allow to follow it properly. Response: Thank you for your comments. I’m sorry for the poor writing, we take it seriously and did a major revision to the manuscript according to the reviewers’ comments. We hope this revision can make our manuscript more acceptable.
Reviewer 2 Report
Dear authors,
I have revised your manuscript. Your idea of using soybean dregs and thus avoiding waste is very good. However, your manuscript requires improvement. Inside the file you will find several comments.
Likewise, I recommend avoiding sentences in which you only mention what an author did (XXXXX [1] found...), without there being a conclusion of what you want to convey.
Finally, I recommend an English editor with the aim of improving grammar and writing.
Best regards

Author Response
Dear Reviewer: Thank you for your comments concerning our manuscript entitled “Wet Fermented Soybean Dregs and Corn Stover Pelleting for Ruminants Feeds” (ID: animals-1788697). We quite appreciate your favorite consideration and the your insightful comments. We did a major revision to the manuscript according to your helpful comments. We hope this revision can make our manuscript more acceptable. Revised portion are marked in blue in the manuscript. The revisions were addressed point by point below. Please feel free to contact us if there are any questions. Sincerely yours, Response to the reviewer’s comments: I have revised your manuscript. Your idea of using soybean dregs and thus avoiding waste is very good. However, your manuscript requires improvement. Inside the file you will find several comments.Likewise, I recommend avoiding sentences in which you only mention what an author did (XXXXX [1] found...), without there being a conclusion of what you want to convey.Finally, I recommend an English editor with the aim of improving grammar and writing. Response: Thank you very much for your comments. We did a major revision to the manuscript according to your comments, and carefully checked and revised the grammar and writing. 1. Title needs to be more precise. Consider that there were no evaluations in ruminal fermentation variables or in the productive performance of the animals. Therefore, the term "ruminants" should be deleted. Focus the title on the physical characteristics of the pellet. Response: Thank you for your comments. We revised the title of the manuscript into “Effects of Wet Fermented Soybean Dregs on Physical and Mechanical Properties of Pellets of Corn Stover” according to your suggestion. 2. It is necessary to link the previous idea with the following one. It could be mentioned that corn stover is used in different ways in cattle feed, including pelleting. Response: Thank you very much for your suggestions. The feed utilization ways of corn stover were introduced in the manuscript according to your suggestion. 3. Specify that they are feed for livestock, specifically ruminants. Briefly mention why corn stover is used in ruminants. Response: Thank you for your valuable comments. The research of corn stover used as feed livestock, specifically ruminants were introduced according to your suggestion. 4. Rather than individually describing each of the studies, it should be mentioned how the additives improve the characteristics of the pellet. Likewise, mention what are the additives used in the pelletization. Response: Thank you for your valuable comments. The effects of existing additives on of characteristics straw pellets are summarized and described in the manuscript. 5. It is necessary to mention that soybean dregs contain anti-nutritional factors, which can decrease with a fermentation process. Include what are the antinutritional factors, briefly describe the health problems in animals and the fermentation process. Response: Thank you for your valuable comments. The effect of fermentation on the anti-nutritive factors (trypsin inhibitor, phytic acid, tannin) of soybean dregs was introduced in detail in the manuscript according to your comment. 6. Check that this statement is true. Compare the crude protein content of soybean dregs, soybean meal, and other sources. Use the basis. Response: Thank you very much for your suggestions. Existing studies [1-3] also showed that soybean dregs can replace soybean meal and other protein feeds. So we listed existing research to support the statement "Fermented soybean dregs can be easily digested and absorbed by animals and it is a substitute of soybean meal and other protein feeds" according to your comment. Ref: [1] Tong, D. StudyonFermentationMethodofSoybeanResidueand ApplicationinBeefCattleFeed. Huhhot: Inner Mongolia Minzu University, 2020.7 [2] Harthan, L.B.; Cherney, D.J.R. Okara as a protein supplement affects feed intake and milk composition of ewes and growth performance of lambs. Animal Nutrition 2017, 3, 171-174. doi:10.1016/j.aninu.2017.04.001. [3] Zang, Y.; Santana, R.; Moura, D.C.; Galvão Jr., J.G.B.; Brito, A.F. Replacing soybean meal with okara meal: Effects on production, milk fatty acid and plasma amino acid profile, and nutrient utilization in dairy cows. Journal of Dairy Science 2020, 104, 3109-3122. doi:10.3168/jds.2020-19182. 7. Mention if soybean dregs have been used in cattle feed and describe, in a general way, what the results were. If not, mention some other soybean by-products. Response: Thank you for your valuable comments. The research of soybean dregs used in feed was introduced in the manuscript according to your suggestion. 8. This statement needs to be verified and should be removed. Response: Thank you very much for your suggestions. This statement (Lines 90 to 91) has been removed in the manuscript. 9. Straw or corn stover? Use the appropriate term. Response: Thank you very much for your careful revision. We are very sorry for that the terms are not unified in the manuscript. Straw was changed to corn stover in the manuscript. 10. This statement seems more like a conclusion than part of the introduction. Response: Thank you very much for your valuable comments. This statement (Line 96 to 98) has been removed in the manuscript. 11. I recommend giving more details of the origin of the materials, including location. Response: Thank you very much for your suggestions. The details of the origin of the materials were described in the manuscript. 12. What do you mean with sampling? Response: Thank you very much for your valuable comments. We are very sorry for the writing mistake in the manuscript. “sampling” is replaced by “Materials treatments”. 13. Better writing is required to understand which combinations were tested. Response: Thank you very much for your valuable comments. We are very sorry for that the description of the experimental treatments was not clear. The experimental treatments were described in detail in the manuscript according to your comments. 14. What do you mean with "copies". Response: Thank you very much for your valuable comments. We are very sorry for the writing mistake in the manuscript. “copies” is replaced by “samples”. 15. What kind of testing? Response: Thank you for your comments. The samples were used as pelleting test. It was revised in the manuscript. 16. Do you mean "corn stover"? Response: Thank you very much for your suggestions. We are very sorry for the writing mistake in the manuscript. “corn stalks” is replaced by “corn stover”. 17. It is not necessary to include a photo of the apparatus. It could be considered as supplementary material. There, each part can be described in detail. Response: Thank you very much for your valuable comments. Indeed, as you say, photographs are not necessary, so the photo of the apparatus have been removed. 18. Eliminate. A figure is meaningless if it is not described within the manuscript. Also, there should be a more detailed description in the footnote. Response: Thank you very much for your suggestions. In fact, Figure 2 shows only part of the feed pellets. Figure 2 has been removed according to your comment. 19. There is no necessary to include this phrase. Consider this suggestion throughout the manuscript. Response: Thank you very much for your suggestions. This sentence (Line 172, 203, 240, 277, 309, 344 ) has been removed throughout the manuscript. 20. That's a lot of references for such a short discussion. Eliminate some and discuss better with the results found in previous articles. Response: Thank you for your comments. The some references was removed according to your suggestion. 21. What columns are you referring to? There are no columns in the figure. Consider this comment in the other figures. Response: Thank you very much for your suggestions. We are very sorry for the writing mistake in the manuscript. We have changed columns to broken line, and it was corrected throughout the manuscript. 22. Include a reference. Response: Thank you very much for your suggestions. We have added a reference in the manuscript ([1]). Ref: [1] Chen, T.Y; Jia, H.L; Zhang, S.W; Sun, X.M; Song, Y.Q; Yuan, H.F. Optimization of Cold Pressing Process Parameters of Chopped Corn Straws for Fuel. Energies 2020,13(3). doi:10.3390/en13030652. 23. Only one of the four studies described here used soybean dregs. Many studies have evaluated different forms and by-products of soybeans, which does not indicate that it is feasible to use fermented soybean dregs as part of the diet of ruminants Response: Thank you very much for your valuable comments. The last paragraph in result and discussion section (L377-387) have been removed according to your suggestion.
Reviewer 3 Report
The purpose of this study was to evaluate the effects of inclusion different levels of wet fermented soybean dregs to ground corn stover as additive to improve pelleting process.
This manuscript reports a topic pertinent to contemporary. The argument to justify the study is strong, the manuscript is well written and organized. The statistical analyses are adequate and report a quality data. Discussions are properly sustained, but in my opinion, there is very little flaws which should be rectified before publication and the one that I expose below.
L74-75: It is necessary to rearrange the statements provided from L74 to L89 as follows. Please, to sustain the statement “"Fermented soybean dregs can be easily digested and absorbed by animals and are a substitute of soybean meal and other protein feeds" insert immediately the paragraph from L79 to L88. (i.e.): In such a way that, Woo et al [24] reported…Tong [25] reported that...Hartan et al [26] reported that the okara...Zang et al [27] showed that..Rahman et al [28] reported that the goats..than those fed commercial pellets. (ending with): Nonetheless, inappropriate storage...[20-21]. So far, only 10% can be...severe wasting [22,23].
L90: “can” is repeated
L109” copies? Or replicas? Or samples? Please clarify
L116: Please, after mention the added levels of soybean dregs, justify the inclusion levels as follows: The levels of soybean dregs tested were considering based on previous reports in which tested inclusion level of soybean residues to ruminant diets were from 5 to 24% without negative effects on DM intake or productivity [Cao et al; Callegaro et al; Zanine et al; Durman et al]. Thus, it can be expected that the additional mass of soybean dregs tested would not affect the needs of animals fed the soybean dregs-corn stover pellets under the better pellet forming conditions.
In such way that the last paragraph in result and discussion section (L377-387) must be removed, since is leftover.
Author Response
Dear Reviewer: Thank you for your comments concerning our manuscript entitled “Wet Fermented Soybean Dregs and Corn Stover Pelleting for Ruminants Feeds” (ID: animals-1788697). We quite appreciate your favorite consideration and the your insightful comments. We did a major revision to the manuscript according to your helpful comments. We hope this revision can make our manuscript more acceptable. Revised portion are marked in blue in the manuscript. The revisions were addressed point by point below. Please feel free to contact us if there are any questions. Sincerely yours, Response to the reviewer’s comments: The purpose of this study was to evaluate the effects of inclusion different levels of wet fermented soybean dregs to ground corn stover as additive to improve pelleting process. This manuscript reports a topic pertinent to contemporary. The argument to justify the study is strong, the manuscript is well written and organized. The statistical analyses are adequate and report a quality data. Discussions are properly sustained, but in my opinion, there is very little flaws which should be rectified before publication and the one that I expose below. Response: Thank you very much for your comments. 1. L74-75: It is necessary to rearrange the statements provided from L74 to L89 as follows. Please, to sustain the statement “"Fermented soybean dregs can be easily digested and absorbed by animals and are a substitute of soybean meal and other protein feeds" insert immediately the paragraph from L79 to L88. (i.e.): In such a way that, Woo et al [24] reported…Tong [25] reported that...Hartan et al [26] reported that the okara...Zang et al [27] showed that..Rahman et al [28] reported that the goats..than those fed commercial pellets. (ending with): Nonetheless, inappropriate storage...[20-21]. So far, only 10% can be...severe wasting [22,23]. Response: Thank you very much for your suggestions. We have revised this paragraph in the manuscript according to your suggestion. 2. L90: “can” is repeated. Response: Thank you very much for your careful revision. “can” has been removed in the manuscript. 3. L109” copies? Or replicas? Or samples? Please clarify. Response: Thank you very much for your suggestions. “copies” is replaced by “samples” in the manuscript. 4.L116: Please, after mention the added levels of soybean dregs, justify the inclusion levels as follows: The levels of soybean dregs tested were considering based on previous reports in which tested inclusion level of soybean residues to ruminant diets were from 5 to 24% without negative effects on DM intake or productivity [Cao et al; Callegaro et al; Zanine et al; Durman et al]. Thus, it can be expected that the additional mass of soybean dregs tested would not affect the needs of animals fed the soybean dregs-corn stover pellets under the better pellet forming conditions. In such way that the last paragraph in result and discussion section (L377-387) must be removed, since is leftover. Response: Thank you very much for your valuable comments. It is revised in the manuscript in blue according to your suggestion, and the last paragraph in result and discussion section (L377-387) has been removed.
Round 2
Reviewer 2 Report
Dear authors,
I appreciate your reply and your time in responding to comments and suggestions. In the file you will find some questions and recommendations.
Check the spaces between words.
Regards.

Author Response
Dear Reviewer: Thank you for your comments concerning our manuscript entitled “Effects of Wet Fermented Soybean Dregs on Physical and Mechanical Properties of Pellets of Corn Stover” (ID: animals-1788697). We quite appreciate your favorite consideration and the your insightful comments. We did a minor revision to the manuscript according to your helpful comments. We hope this revision can make our manuscript more acceptable. Revised portion are marked in blue in the manuscript. The revisions were addressed point by point below. Please feel free to contact us if there are any questions. Sincerely yours, Response to the reviewer’s comments: Dear authors, I appreciate your reply and your time in responding to comments and suggestions. In the file you will find some questions and recommendations. Check the spaces between words. Regards. Response: Thank you very much for agreeing with my revisions and again commenting on the manuscript. We checked and revised carefully the manuscript. 1. What does mean "EM"? Response: Thank you for your comments. "EM" refers to effective microorganisms. The spelling of EM and the manufacturers of yeast starter were added in the manuscript. 2. I recommend removing the "/%" symbols, putting a superscript and putting a footnote with the corresponding description. Evenly distribute the columns. Response: Thank you very much for your valuable comment. The "/%" was removed, and a superscript and a footnote with the corresponding description were added in the manuscript according to your suggestion. 3. What do you mean with "broken line"? Figure? Consider this for the other figures. Response: Thank you for your suggestions. "broken line" means the line chart, it was revised in the manuscript. 4. Focus the conclusion on the best result obtained with the fermented soybean dregs and the particle size of the corn stover. Response: Thank you for your valuable comments. The conclusion on the effect the particle size of the corn stover on physical and mechanical properties of pellets was added in the manuscript according to your suggestion. 5. I would recommend placing this phrase (Line 282 to 285) in the results section. Just mention what is the best percentage of fermented soy bean dregs. Response: Thank you for your valuable comments. This statement (Line 282 to 285) was simplified and placed in the results section according to your comment.
